# The Mutagenic Plasticity of the Cholera Toxin B-Subunit Surface Residues: Stability and Affinity

**DOI:** 10.3390/toxins16030133

**Published:** 2024-03-04

**Authors:** Cheuk W. Au, Iain Manfield, Michael E. Webb, Emanuele Paci, W. Bruce Turnbull, James F. Ross

**Affiliations:** 1School of Molecular and Cellular Biology, University of Leeds, Leeds LS2 9JT, UK; 2Astbury Centre for Structural Molecular Biology, University of Leeds, Leeds LS2 9JT, UK; 3School of Chemistry, University of Leeds, Leeds LS2 9JT, UK; 4Dipartimento di Fisica e Astronomia “Augusto Righi”, Viale Berti Pichat 6/2, 40127 Bologna, Italy

**Keywords:** cholera toxin B-subunit, CTB, GM1 ganglioside, melting temperature, dissociation constant, Rosetta, mutational-space map, differential scanning fluorimetry, isothermal titration calorimetry

## Abstract

Mastering selective molecule trafficking across human cell membranes poses a formidable challenge in healthcare biotechnology while offering the prospect of breakthroughs in drug delivery, gene therapy, and diagnostic imaging. The cholera toxin B-subunit (CTB) has the potential to be a useful cargo transporter for these applications. CTB is a robust protein that is amenable to reengineering for diverse applications; however, protein redesign has mostly focused on modifications of the N- and C-termini of the protein. Exploiting the full power of rational redesign requires a detailed understanding of the contributions of the surface residues to protein stability and binding activity. Here, we employed Rosetta-based computational saturation scans on 58 surface residues of CTB, including the GM1 binding site, to analyze both ligand-bound and ligand-free structures to decipher mutational effects on protein stability and GM1 affinity. Complimentary experimental results from differential scanning fluorimetry and isothermal titration calorimetry provided melting temperatures and GM1 binding affinities for 40 alanine mutants among these positions. The results showed that CTB can accommodate diverse mutations while maintaining its stability and ligand binding affinity. These mutations could potentially allow modification of the oligosaccharide binding specificity to change its cellular targeting, alter the B-subunit intracellular routing, or impact its shelf-life and in vivo half-life through changes to protein stability. We anticipate that the mutational space maps presented here will serve as a cornerstone for future CTB redesigns, paving the way for the development of innovative biotechnological tools.

## 1. Introduction

In the coming era of personalized medicine, the ability to move theragnostic agents across the mammalian cell membrane will be of great importance to biotechnology and healthcare. Previously, various strategies for membrane trafficking have been explored including cell-penetrating peptides [1], liposomes [2], viral vectors [3], nanoparticles [4], and bacterial toxins [5]. One well-used trafficking protein is the cholera toxin B-subunit (CTB) [6], a member of the expansive AB_5_ toxin family [7]. These toxins gain access to the cell interior through binding to specific gangliosides present on a subset of cells, granting cellular specificity to the trafficking mechanism, thus having the potential to reduce off-target effects of any accompanying cargo molecule.

CTB is a member of the AB_5_ family of bacterial toxins characterized by their toxic A-subunit (Figure 1A) and pentameric trafficking B-subunit (Figure 1B) [8]. CTB is most often used as an adjuvant for immunological studies [9], or for trafficking cargoes into cell interiors, such as fluorophores [10], nanoparticles [6], biologics [11], and antigens [12]. CTB is a structurally robust protein, making it resistant to modification, thus CTB has already been used in a number of bioengineering and synthetic biology pursuits, such as assembly into higher order geometries [13], mediating membrane fusion [14], and labeled via sortase ligation [15]. The cholera toxin (CT) (Figure 1A) trafficking pathway initiates when CTB binds to glycans presented on the cell surface [16,17,18], including ganglioside GM1 (shown in black in Figure 1C), triggering endocytosis [19] and retrograde trafficking. CTB alone has been observed to traffic to the Golgi apparatus [20], where it accumulates in the medial compartment [21]. The presence of a KDEL tag on the cholera toxin A2-subunit (CTA2) facilitates retrograde trafficking to the endoplasmic reticulum (ER), but mutations of this sequence also reach the ER, albeit with reduced efficiency [22]. In the ER, the CTA1 subunit is released via disulfide-bond cleavage. Now free in the ER lumen, CTA1 is recognized as being a misfolded protein and retro-translocated into the cytoplasm to mediate its toxic effect [23]. Various cell types, including motor neurons, display the high-affinity GM1 ligand for CTB on their surfaces [24,25], making CTB a viable candidate delivery vehicle for these types of cells [26]. However, there are homologous proteins to CTB that bind a range of other gangliosides. 

Gangliosides, cerebrosides, and globosides are complex glycosphingolipids abundantly present on mammalian cell surfaces, contributing to the glycocalyx [27,28]. Their composition varies among cell types and tissues and during processes like the cell cycle, forming the ‘Glyco-code’, a cell-type and cell-health-specific extracellular flag [29,30,31]. Exploiting these biomarkers is attracting interest from drug delivery and immunomodulatory medicine [32]. Prominent gangliosides, including GM1, GD2, and GM3, exhibit distinct expression patterns, associations with diseases, and implications in neurodegenerative disorders and cancer [33,34,35]. CTB and other AB5 toxins like heat-labile enterotoxin, Shiga toxin, and pertussis toxin, show promise as tools for drug delivery due to their selective binding of these glycolipids; however, modulation of their binding sites for association with non-native gangliosides is required for new diagnostic and therapeutic applications. Reengineering these toxins to bind non-native gangliosides could broaden their cell-targeting repertoire, offering new avenues for diagnostic and therapeutic technologies.

As such, CTB has been subjected to a number of mutational studies to monitor the effect of mutations on the toxicity and structural robustness of the protein. Figure 1D summarizes these mutations in a curated mutational-space map, highlighting mutations that maintain toxicity, evidence of GM1 binding, evidence of pentamer formation, and no evidence of expression (it can be assumed that GM1 binding requires pentamer formation and toxicity requires GM1 binding). Much of the data in Figure 1D are from the work of Jobling and Holmes from 1991 and 2002 [36,37]. The first of these papers analyses four GM1 binding residues, positions 33, 34, 35, and 88, by saturation mutagenesis, characterizing all successfully expressed mutants at those positions (and assuming those not seen to be non-expressible), as well as two Cys-Ser mutations at positions 9 and 86. The second of the two papers uses directed mutagenesis to generate a series of mutations across the protein. Further studies from the groups of Hirst, Holmes, and Bernardi add a selection of characterized mutants at additional sites [38,39,40]. Figure 1E summarizes the mutational diversity of 254 sequences returned from a protein-BLAST of CTB (classical biotype). Among the sequences returned were 119 sequences of CTB, 86 sequences of LTB (heat labile enterotoxin type I, with 80% sequence identity to CTB) [41], 26 sequences of CfxB (*C. freundii* AB5 toxin, 73% sequence identity to CTB) [42], and 23 sequences of EcxAB (*E. coli* AB5 toxin with the metalloprotease A-subunit, sharing 63% sequence identity to CTB) [43]. These propensities are scaled from orange to blue, indicating a probability of 0 to 1, with white for zero occupancy. It is worth stressing that these propensities are directly from a BLAST search and are provided to show the observed mutational variation across the four variants described above within that search only. This natural variation in close structural homologs combined with the curated mutational data gives a detailed overview of the current mutational diversity observed for CTB.

In this work, we investigated further the mutagenic plasticity of the surface residues of CTB to aid future protein engineering studies. We performed a computational saturation scan measuring stability across all surface residues of CTB, excluding the central pore where the A2-linker peptide binds (Figure 2, cyan), accompanied by an experimental alanine scan at 40 selected positions. The alanine scan mutants were characterized by differential scanning fluorimetry (DSF) to determine their melting temperature (T_m_). Where these positions include GM1 binding site residues (Figure 2, magenta), we have determined the GM1 oligosaccharide binding affinity by isothermal titration calorimetry (ITC) [44] and their computational interaction energies. It was not practical to measure changes in the binding affinities of the mutant proteins to the Lewis-y glycan as the wild-type protein binds this glycan with a very low affinity [45]. 

## 2. Results

To assess the changes in stability and affinity upon mutation, a set of computational saturation mutagenesis scans were performed using the Rosetta protein design software package [47,48]. The Rosetta protein design suite generates conformational ensembles of protein structures through Monte Carlo sampling and assesses protein stability through an internal all-atom force field, providing a score in Rosetta Energy Units (REU), akin to kcal/mol. For all computational analyses, a symmetrized CTB pentamer was generated from the Classical biotype CTB structure 2CHB from the protein data bank [46] both in the presence and absence of the GM1 oligosaccharide. The initial model was relaxed in Rosetta 1000 times and the lowest energy model was used to initiate further analysis. Due to the pentameric nature of CTB, mutations were necessarily replicated on each monomer of the subunit simultaneously, effectively reinforcing the effect of the mutation five-fold. The change in Rosetta Energy Units (ΔREU) scoring function was calculated by subtracting the mutant REU from the wild-type REU (ΔREU = REU_mut_ − REU_wt_). For stability calculations, the total score was used, and for the interaction calculations, the difference between ligand-bound and unbound REU was determined.

### 2.1. Computational Saturation Scan—CTB Ligand-Free Structure

A computational saturation scan was performed on the CTB structure in the absence of its bound GM1 oligosaccharide ligand (number of repeats = 100). A saturation mutagenesis scan was performed for each of the positions highlighted in Figure 2, in which the wild-type residue was changed one by one to each of the 20 canonical amino acids. The ΔREU values (vs. wild-type CTB) are displayed in Figure 3A. The color scale in Figure 3A ranges from ΔREU −31 (most blue = most stabilizing) to ΔREU ≥ +100 (most red = most destabilizing). A ΔREU of zero is displayed as white and the wild-type residue is highlighted with a gold border. The standard deviation for wild type-to-wild type (wt-to-wt) mutations is 1.1 REU, with a minimum and maximum of −2.7 and +3.2 REU, respectively. A more thorough alanine scan was also performed (repeats = 1000), which can be seen in Appendix A; however, the difference in medial average between the 100 repeat and the 1000 repeat experiments was negligible (Appendix A). 

As expected, mutations to proline had the most disruptive effect on the protein structure, as most residues’ backbone geometries do not conform to the phi angle restrictions imposed by the proline sidechain. But for two residues G33 and H94, proline mutations generate the best stabilizing effect among all mutations, presumably by rigidifying these residues’ native backbone conformations. As would also be expected, the majority of mutations across the saturation scan were destabilizing; however, there were some notable structural intricacies. N14, N21, K34, G45, and E79 were five of the top six positions most destabilized by proline substitutions, and they were also destabilized by beta-carbon branched amino acids, notably valine (the top 5/5 of its destabilizing substitutions) and isoleucine (the top 5/6 of its most destabilizing substitutions), and also including threonine (the top 5/9 of its most destabilizing substitutions). Indeed, on inspection, these residues uniquely have a positive phi angle (Appendix A), generating steric clashes with any introduced branched beta-carbons as well as introducing strain to the protein backbone. 

Counter to this, there were also residues that showed a preference for branched beta-carbons, specifically T19, T28, and T47 (where T47I is a signature of the El-Tor biotype), and to a lesser extent H18 and N89. At most positions, the wild-type residue was the lowest energy amino acid; however, there were a number of positions where the wild-type residue was (one of) the highest energy options, namely T1, G33, I58, and S60, and to a lesser extent H13, G54, and Q56 (for hydrophilic residues only). Notably, with the exception of the N-terminal residue T1, these residues all reside in flexible loops around the GM1 binding site, with H13, G33, Q56, and I58 in direct proximity to the ligand. Presumably, these residues sacrifice structural stability for increased ligand binding affinity and specificity by directly increasing the enthalpy of interaction or providing flexibility to the region to facilitate binding. Alternatively, residues responsible for Lewis-y binding (Q3, Q16, H18, NGATF 44–48, N89, and KTPH 91–94) [49] show limited stability increases upon mutation, with the only notable stability increase being with Q3T, H18P/V/I, T47V/I, and T92D/N. The only remaining significant stability increases were with T6D, F25K, T28V/I, K43G, E51P, S55P, R58E/D, H94P, and A102I.

### 2.2. Computational Saturation Scan—CTB Ligand-Bound Structure

A second saturation mutagenesis scan was performed with the GM1 ligand present. In this experiment, the ΔREU stability was determined as previously (Figure 3B), but additionally, an interaction energy was calculated between the protein and ligand (Figure 3C). Comparing the stability, on a residue-by-residue basis, of the ligand-free structure (Figure 3A) with the stability of the ligand-bound structure (Figure 3B) it can be seen that many of the stability increases gained upon mutation without the ligand present are lost when the ligand is present. Most notably, H13, G33, K34, and Q56 lose all stability increases, with the exception of G33A and K34G. Contrastingly, I58 maintains the character of stability increases across several of the ligand-free and ligand-bound structure mutations. Of potential interest, a number of mutations at K91 appear to increase the stability of the ligand-bound structure relative to the ligand-free structure. Changes to the interaction energy can be seen in Figure 3C, where notably there are no mutations that increase the interaction energy across residues E11, Y12, R35, E51 (bar E51Q), or W88 (bar W88F). There are a few positions where interaction energies can be improved, such as at N14, G33, H57, I58, and Q61; however, these mutations largely sacrifice stability for increased interaction, with the exceptions of G33A, K34G, K91H, and a selection of mutations at I58 (EDQM). It is perhaps unsurprising that this investigation only found 7 mutations among 320 that increased both the stability and the interaction energy, as such single point mutations can be easily sampled evolutionarily, and if beneficial, would likely have been selected for in current strains. Indeed, of these mutations, G33A is examined further in this paper, K34G is shown to maintain infectivity (Figure 1D), and K91H and I58E/D/Q/M have not been observed in vitro or in vivo (Figure 1D,E).

### 2.3. Expression of CTB Alanine Scan Mutants

Alanine mutations of CTB were generated via gene synthesis, assembly PCR, or site-directed mutagenesis in the plasmid pSAB2.2 described previously [50]. Following gene expression in *E. coli*, ammonium sulfate precipitation and nickel affinity purification were performed to obtain the mutated proteins. Each protein was analyzed for purity and stability by SDS-PAGE with 0.5 nmol of protomer loaded per lane. The protein—loading dye mixture was not boiled prior to electrophoresis to allow intact pentamers to be observed on the gels. Figure 4 shows the lanes for each of the mutant CTB proteins. Differences in electrophoretic motility can be seen for a number of mutants, as can the concentration of pentameric protein, which remained intact during SDS-PAGE. The gel image was cropped just before the dye front as in some cases the dye front and band for CTB monomers were coincident, leading to ambiguity over the amount of protomer. The original gels are available in Appendix A. It should also be noted here that CTB pentamers are obligate multimers and purification of CTB via nickel affinity chromatography is dependent on having at least five surface histidine residues arising from pentamerization. A notable absentee from the expressed proteins is H13A, which could not be isolated by nickel affinity chromatography, and is thought to be one of the residues primarily responsible for nickel binding [51]. All other proteins were expressed to a reasonable degree; however, the pentameric stability on SDS-PAGE varied significantly across the mutants. The melting temperature of these proteins was then assessed by DSF (Figure 5) and where appropriate the GM1 binding affinity was determined by ITC (Figure 6).

### 2.4. Differential Scanning Fluorimetry of Alanine Scan Mutants

To monitor the thermostability of the alanine scan mutants, the melting temperature (Tm) was determined by differential scanning fluorimetry (DSF) in the presence of the dye molecule SYPRO Orange [52]. Figure 5 shows the data from DSF experiments normalized against coincident runs of the wild-type protein (the raw data can be seen in Appendix A). The wild-type protein gave a Tm of 80 °C in line with literature precedents [53], and predictably, most alanine mutations caused a reduction in Tm. In general, this reduction in Tm was modest with 20 of the 40 mutants yielding a Tm between 75 and 79 °C. However, some of the mutants yielded lower melting temperatures, most notably Y12A with a Tm of 49 °C, which was also predicted to be one of the most destabilizing mutations on the computational alanine scan (+28 ΔREU, Appendix A). The hydroxyl group on the side chain of Y12 functions as a hydrogen bond donor to the backbone carbonyl oxygen of G33 (Appendix A). The replacement of Y12 with alanine (Y12A) disrupts these interactions, and the lack of steric bulk exposes the hydrophobic side chains of neighboring residues and the protein core, leading to a reduction in stability. 

Conversely, some Tm values were larger than those for the wild-type protein, most notably G33A, with a Tm of 89 °C, and correspondingly, it had the greatest stability improvement predicted by Rosetta (−15 ΔREU, Appendix A). This negative correlation of decreasing ΔREU and increasing Tm is reasonably representational of the data set, which had an R-squared of 0.45 (Figure 7A) after the removal of three outlying data points (Appendix A). 

While the general trends agree between the calculated and experimentally measured protein stabilities, some differences were observed. These probably arise from differences in estimating the bonded and non-bonded atomic interaction energies compared to thermally melting proteins, as well as experimentally specific variables such as pH and ionic strength. However, the analysis does generate predictions that may reasonably guide future design choices with changes of roughly −2 ΔREU corresponding to a 1 °C increase in Tm. 

Four of the mutants did not yield a melting temperature: S26A, N44A, A46G, and W88A. Of these mutations, S26A and N44A showed lower amounts of intact pentamer after SDS-PAGE (a signifier of reduced stability, Figure 4); however, A46G and W88A had pentameric bands on the SDS-PAGE similar to the wild-type. N90A surprisingly had a significantly larger fluorescence intensity than the other samples, for which we can only muse that the interplay of the dissociation of pentamers and unfolding of the protein is responsible. Indeed, the DSF fluorescence intensity for CTB was quite reduced compared to other proteins given the total amount of protein in solution, and the intensity of N90A was closer to the expected fluorescence for the protein concentration.

### 2.5. Isothermal Titration Calorimetry of Alanine Scan Mutants

To test the effects of the mutations on GM1 binding, isothermal titration calorimetry (ITC) was performed for variants with side chains within 6 Å of the GM1 ligand (positions E11A, Y12A, N14A, G33A, K34A, E51A, Q56A, H57A, I58A, Q61A, W88A, N90A, and K91A; data in Appendix A). The wild-type CTB was found to have a dissociation constant (K_d_) of 43 nM, in line with literature precedents [44] (Figure 6). As with the stability improvements, we did not presume to find single point mutations that increased the affinity, as these would likely already exist among natural strains; rather, our interest is in how these mutations modulate the binding affinity of the native ligand. 

Two mutations, W88A and K91A, caused a complete abrogation of binding. W88 forms a classical CH-π interaction with the terminal galactose residue of GM1, which is common in carbohydrate-binding proteins [54,55], but it also contributes to the stability of the hydrophobic packing of the binding site (Appendix A). Therefore, the loss of the binding affinity with this alanine substitution is unsurprising. However, seemingly equally disruptive is the K91A mutation. This residue forms hydrogen bonds with the GM1 terminal galactose residue and a salt bridge to E51 (Appendix A). It appears that the important interaction is to the GM1 galactose, rather than the salt bridge, as the E51A mutation (removing the same salt bridge) is considerably less disruptive to GM1 binding (with a measured K_d_ of 483 nM). 

It was not possible to determine a K_d_ for three of the other mutants, Y12A, G33A, and H57A, although the thermograms showed evidence for weak interactions as partial binding curves were obtained (Figure 6C and Appendix A). The protein-ligand interaction calculations from the Rosetta saturation mutagenesis scan predicted that the only alanine scan mutants to give improvements in GM1 binding would be G33A and H57A. However, H57A could only achieve this increase in interaction energy by sacrificing overall stability. The DSF data indicated only a marginal reduction in Tm compared to the wild-type protein; however, the ITC results demonstrated a severe loss of GM1 affinity. H57 does not directly bind to GM1, but both its backbone and sidechain form hydrogen bonds with preceding residues in the protein chain. These interactions potentially stabilize positions 51 to 57 that encompass three residues crucial for GM1 binding. Mutating these positions to alanine may increase flexibility in this area, perturbing binding to GM1. This finding contradicts previous results indicating that H57A maintains the ability for GM1 binding, determined through ELISA and surface plasmon resonance [38]. As ITC quantifies a 1:1 binding interaction between a single GM1os and the CTB pentamer, we propose that the avidity effects of ELISA and SPR are responsible for the previously observed binding, or these changes are due to affinity differences between the classical biotype (this study) and the El Tor biotype (ELISA and SPR). 

This leaves G33A as the only predicted beneficial mutation. As G33 exhibits greater torsional flexibility due to the absence of steric hindrance from a beta-carbon, the introduction of a beta-carbon through G33A rigidifies the protein backbone, enhancing stability, as seen in the DSF. However, this also appears to severely reduce binding to GM1, suggesting that either beta-carbon steric hindrance or a loss of conformational flexibility is responsible. Previous studies demonstrated that G33D resulted in a loss of GM1 affinity, with crystallographic studies indicating that aspartic acid obstructs the docking of the sialic acid residue of GM1 [56]. Another plausible explanation is that G33 plays a role in facilitating conformational changes to accommodate GM1 binding and plays a role in increases in strain, as discussed above for N14, N21, K34, G45, and E79, and discussed for Q49 and V50 in the 1.25 Å structure of CTB by Merritt et al. (1998) [57]. 

The remaining mutations were all predicted to decrease the interaction affinity. When the Rosetta interaction energies are compared to the experimental ITC affinities (Figure 7B), an R-squared of 0.51 is observed between the datasets (Appendix A). Similar to the correlation between melting temperature and Rosetta stability, the correlation between affinity and interaction energy allows for reasonable predictions of affinity to be made, with a relationship of −1 REU to a ~−230 nM change in K_d_ observed for the affinity, within the range measured.

## 3. Discussion

This study computationally evaluated the mutagenic plasticity of 58 surface residues of CTB and verified the results with a 40-residue experimental alanine scan, including residues responsible for interacting with the GM1 oligosaccharide. We employed a comprehensive analysis combining computational saturation scans, in the presence and absence of GM1 oligosaccharide, and experimental alanine scans, assessing the melting temperature by DSF and the affinity by ITC.

The computational analyses revealed the impact of mutations on CTB stability, with certain residues around the GM1 binding site demonstrating unexpected stabilizing effects, such as G33A, and others behaving predictably based on their backbone geometry, such as N14, N21, K34, G45, and E79. The alanine scan DSF was used to link the experimental validation of melting temperatures to computationally estimated stability, validating these predictions and showcasing the robustness of CTB as a protein amenable to diverse mutations. Notably, the saturation scan predicted that a selection of mutations at positions 1, 13, 33, 54, 56, and 58 are stabilizing, and may be useful residues to mutate to improve thermostability.

Correspondingly, the ITC experiments for the alanine mutants provided validation for the computationally predicted interaction energies, with all but one of the affinities predicted well. Notably, alanine residues at specific positions, such as W88 and K91, resulted in a complete abrogation of GM1 binding, underscoring the importance of these residues in the interaction; however, the saturation scan predicted that W88F and K91H/F specifically are less deleterious substitutions. Interestingly, Jobling & Holmes (1991) did not observe W88A in their saturation scan [36]; however, we observed reasonably good expression. Nor did they observe W88F. Additionally, the saturation scan predicted that positions 57, 58, and 61 can accept a wide range of mutations that increase the affinity, but with the exception of 58, cause a significant loss of stability.

While the experimental measurements may not have direct physical relevance to computational estimates, an R-squared of ~0.5 for each case provides a reasonable correlation to the experimental data (Figure 7). This correlation adds validity to the saturation mutagenesis scans and their predictions, offering valuable insights. However, this computational technique only considers static mutations and it does not estimate complications from disrupted folding pathways, increased proteolysis, reduced secretion, or other ‘biological’ concerns, and as such, occasional inaccuracies are observed.

This study provides a basic scientific understanding of surface residue plasticity for future CTB redesign, providing a nuanced understanding of the mutational landscape and its implications for stability and ligand binding. The established relationships contribute to a comprehensive guide for researchers aiming to customize CTB for specific functions, such as a super-thermostable CTB by selecting mutants that increase the Tm of CTB or modulation of the GM1 binding interface to either increase its specificity for the native glycan, alter its specificity to a different glycan, broaden its specificity to generate a glycan-promiscuous CTB, or design a CTB that binds to an entirely different class of molecule. Further investigations in this area could focus on mutations that increase the acid stability of the toxin, allowing for improved oral administration of therapeutics, or could focus on investigating the mutational plasticity of other members of the AB5 toxin family. We anticipate the mutational space maps presented here will pave the way for the development of innovative biotechnological tools, capitalizing on CTB’s structural robustness and its potential for accommodating various mutations.

## 4. Materials and Methods

### 4.1. Computational Setup 

All calculations were performed using Rosetta 3.10 [47] on the University of Leeds Arc3 and Arc4 high-performance computers. The Rosetta ‘Fast relax with design’ function was used with the ‘beta’ scoring force field (which includes scoring terms for water bridges and glycosidic linkages [48]). Pdb 2CHB was used as the scaffold structure [46]. From this structure, chain E (protein) and chain B (glycan) were selected, copied, aligned, and used to substitute the original subunits, thereby generating an atomically symmetric starting model for analysis. The chains were renamed and numbered appropriately. For both the ligand-free and ligand-bound structures, this symmetrized CTB molecule was relaxed 1000 times in the presence or absence of the ligand, respectively, and the lowest energy structure was then used as the starting pose for further modeling.

### 4.2. Computational Alanine Scan (Stability) 

In this investigation, we conducted an alanine scan across the 40 experimentally assessed surface residues highlighted in the study. Starting with the model described above, for each position we generated 1000 relaxed structures of the alanine mutant as well as 1000 dummy relaxations of wt-to-wt mutations, generating a total of 80,000 relaxations. As the mutation was applied once per monomer, five mutations were applied per pentamer, totaling 200,000 assessed alanine mutations. The average of the wild-type REU ‘total_score’ was then subtracted from each of the averaged alanine REU ‘total_scores’ to give the mutated ΔREU for each alanine mutation. These are shown as violin plots in Appendix A.

### 4.3. Computational Saturation Scan (Stability)

In this investigation, we conducted a saturation scan across the 58 surface residues highlighted in the study. Starting with the model described above, for each position we generated 100 relaxed structures for each canonical amino acid as well as 100 dummy relaxations of wt-to-wt mutations, generating a total of 84,000 relaxations. As the mutation was applied once per monomer, five mutations were applied per pentamer, totaling 420,000 assessed mutations. To calculate the ΔREU score, we averaged the ‘total_score’ for the 100 poses for each mutation and subtracted the average ‘total_score’ for the 100 ‘wild-type’ poses for that position. We then compared the medial average of the 1000 × alanine scan and the average from the 100 × saturation scan to confirm appropriate sampling in the saturation scan (Appendix A).

### 4.4. Computational Saturation Scan (Ligand Interaction)

In this investigation, we conducted a ligand-bound saturation scan across the 13 GM1 interface residues highlighted in the study. The process for determining the stability and the interaction energy for each mutation in the presence of the ligand was identical to that described above for the stability calculation in the absence of the ligand, with the exception that when the ligand was present, in addition, the interaction energy was calculated using the ‘InterfaceAnalyzer’ of Rosetta 3.10 on the mutated structures.

### 4.5. Molecular Biology

The plasmid vector pSAB2.2 was used as the expression system for *E. coli* to generate the native and alanine scan CTB molecules. pSAB2.2 is a derivative of pMAL-p5x (New England Biolabs, Ipswich, MA, USA), described previously [50], containing the LT-IIb periplasmic leader sequence. Alanine mutations at non-GM1 interface positions were generated by Assembly PCR, as described in Ross et al., 2019 [13]. Primers for assembly PCR were obtained from Integrated DNA Technologies. GM1 interface position mutations were ordered as whole genes (Twist Biosciences, San Francisco, CA, USA) and amplified using Platinum™ PCR SuperMix High Fidelity (Thermo Fisher Scientific, Oxford, UK). In both cases, assembled or amplified genes and the pSAB2.2 plasmid were digested using SphI-HF and HindIII-HF. The plasmid digest reaction included Quick CIP (all New England Biolabs) in accordance with the manufacturer’s instructions. The pSAB2.2 digest was gel extracted (Gel Extraction Kit, Qiagen, Manchester, UK) and the gene digests were heat inactivated at 80 °C. Ligation was performed in accordance with the manufacturer’s instructions in 10 µL for 1 h at room temperature using T4 DNA ligase (New England Biolabs). Transformations were conducted with 3 µL of ligation mixture and 10 µL of XL1-Blue or XL10 Gold (Agilent, Santa Clara, CA, USA), and single colonies were isolated and confirmed with sequencing (GeneWiz and GATC). Purified plasmids were subsequently transformed into BL21 (DE3) cells for expression.

### 4.6. Gene Expression and Protein Purification

Protein expression was initiated by growing a 5 mL overnight culture of transformed BL21 (DE3) cells. Then, 1.2 mL of the overnight culture was used to inoculate 400 mL of LB media (100 µg/mL carbenicillin), which was incubated at 37 °C and 200 RPM until an OD_600_ of 0.6–0.8 was reached. At this point, expression was induced with IPTG (1 mM final), and the culture was incubated at 30 °C and 200 RPM overnight. Fractionation was conducted at either 4k× *g* for 20 min or 10k× *g* for 10 min. The pellet was isolated and frozen at −80 °C for long-term storage (typically approximately 50% of the protein yield can be found in the cell pellet (periplasmically located). The clarified media (400 mL) was adjusted to 57% ammonium sulfate slowly while stirring and incubated at RT, stirred for 2 h, before centrifugation at either 4k× *g* for 40 min or 17k× *g* for 20 min. The precipitated protein was resuspended in Ni-Wash Buffer (100 mM Tris, pH 8.0, 50 mM NaCl, 20 mM imidazole), clarified through 0.4 and 0.22 µm syringe filters, and applied to Ni-NTA agarose beads (HIS-Select, Millipore, Feltham, UK) in a gravity drip column. After washing with 5 CV of Ni-Wash Buffer, elution was performed with 3 CV of Ni-Elution Buffer (100 mM Tris, pH 8.0, 250 mM NaCl, 500 mM imidazole), isolated in 1 mL fractions. Fractions were analyzed by SDS-PAGE, pooled based on purity, and stored at −20 °C. Before experimental evaluation, the samples underwent dialysis against 2 L Tris buffer (100 mM Tris, pH 8.0, 250 mM NaCl) to remove the imidazole component. Subsequently, the samples were concentrated to 80 µM.

### 4.7. Sodium Dodecyl-Sulfate Polyacrylamide Gel Electrophoresis 

Sodium dodecyl-sulfate polyacrylamide gel electrophoresis (SDS-PAGE) was used to evaluate both the purity and the structural integrity of the CTB variants. To visualize the CTB protomers on the gel, standard SDS-PAGE sample preparation was used. First, 0.5 nmol of protein was mixed with 7 µL of NuPAGE LDS Sample Buffer (Invitrogen, Waltham, MA, USA) to a volume of 25 µL and heated to 70 °C for 5 min, vortexed, and spun down. To visualize the CTB pentamers on the gel, the sample preparation heating step was skipped. Electrophoresis was conducted on a polyacrylamide gel with a 15% acrylamide resolving gel and a 5% acrylamide stacking gel. Samples were loaded alongside the Unstained Protein Standard, Broad Range (New England Biolabs) and electrophoresis was conducted at 180 V for approximately 40 min or until the dye front reached the gel bottom. Gels were stained in InstantBlue (Abcam, Waltham, MA, USA) and destained in dH_2_O. Images were captured using an Amersham Imager 600 (GE Healthcare, Hatfield, UK).

### 4.8. Differential Scanning Fluorimetry

For this experiment, 96-well-plates were prepared to 25 µL with a final concentration of 80 µM of CTB and 10 × SYPRO orange per well. Subsequently, the plates were loaded into a quantitative PCR machine (QuantStudioTM3, Thermo Fisher Scientific). The temperature-programmed protocol commenced at an initial temperature of 25 °C, followed by 75 cycles of 8 s, with 1 °C temperature increments per cycle. Emission data were captured at a wavelength of 570 nm. The results were analyzed using an in-house Python script, calculating the differential to find the melting temperature (Tm) of the samples at the peak maximum. The signal-to-noise ratios for S26A, N44A, A46G, and W88A were deemed too low for accurate Tm determination and were flagged as not discernible (N.D.).

### 4.9. Isothermal Titration Calorimetry

ITC experiments were conducted on a MicroCal iTC200 (Malvern Panalytical, Royston, UK) at a temperature of 25 °C in Tris buffer (100 mM Tris, pH 8.0, 250 mM NaCl). The sample cell was filled with CTB variants at 10 µM. GM1os (Biosynth Ltd., Compton, UK) was dissolved in CTB dialysis buffer to a concentration of 110 µM. The ITC run comprised a single 0.5 μL injection of GM1 followed by 19 injections of 2 μL at 2 min intervals. The initial injection data point was excluded from subsequent analysis to account for its role in pre-titration equilibration. The data were processed by subtracting the heat of dilution from a titration of GM1 into the buffer and curve fitting was performed using Microcal Origin software and the Microcal one-site model.

## Figures and Tables

**Figure 1 toxins-16-00133-f001:**
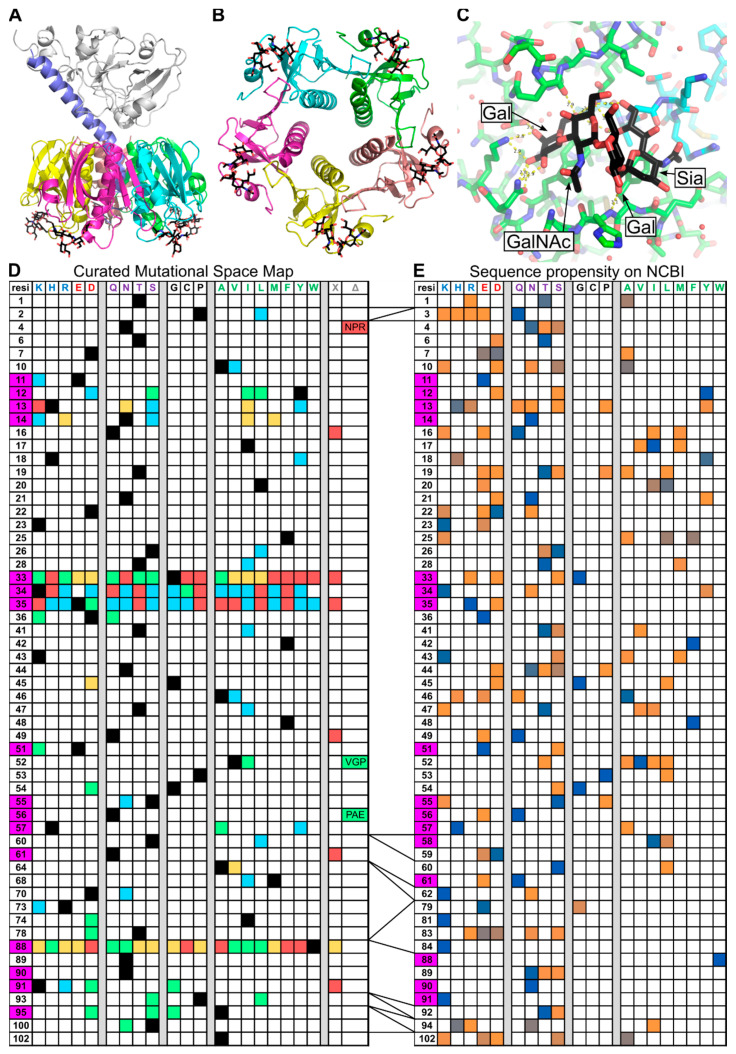
Overview of cholera toxin B subunit structure and sequence. Cholera toxin A1 subunit (grey), A2 subunit (mauve), B subunit (multi-colored), and GM1 oligosaccharide (black). (**A**) View of whole cholera toxin. (**B**) Five-fold axis view of CTB, showing GM1 binding locations. (**C**) Focus on GM1 binding site, involving two CTB subunits and GM1 oligosaccharide. (**D**) Mutations of CTB characterized in the literature, residue positions in left-hand column, wt (black), biologically active (blue), binds GM1 (green), forms pentamer (orange), no expression (red), untested (white), X = termination, Δ = insertion, GM1 binding residues highlighted in magenta. (**E**) Selected surface residue amino acid propensities for 254 sequences from NCBI Blast of CTB sequence (classical biotype), results include sequences returned for CTB, LTB, CfxB, and EcxAB. Color scales from 0 to 1 respective probability (orange—less frequent to blue—most frequent), no data (white), GM1 binding residues highlighted in magenta.

**Figure 2 toxins-16-00133-f002:**
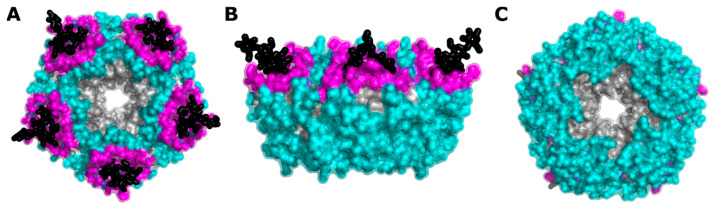
Coverage of cholera toxin B subunit (CTB) mutations in this study. Highlighting the GM1 ganglioside (black), mutated binding site residues (magenta), mutated surface residues (cyan), and non-mutated residues (gray). (**A**) Top view, (**B**) side view, (**C**) bottom view. From pdb:2CHB [46].

**Figure 3 toxins-16-00133-f003:**
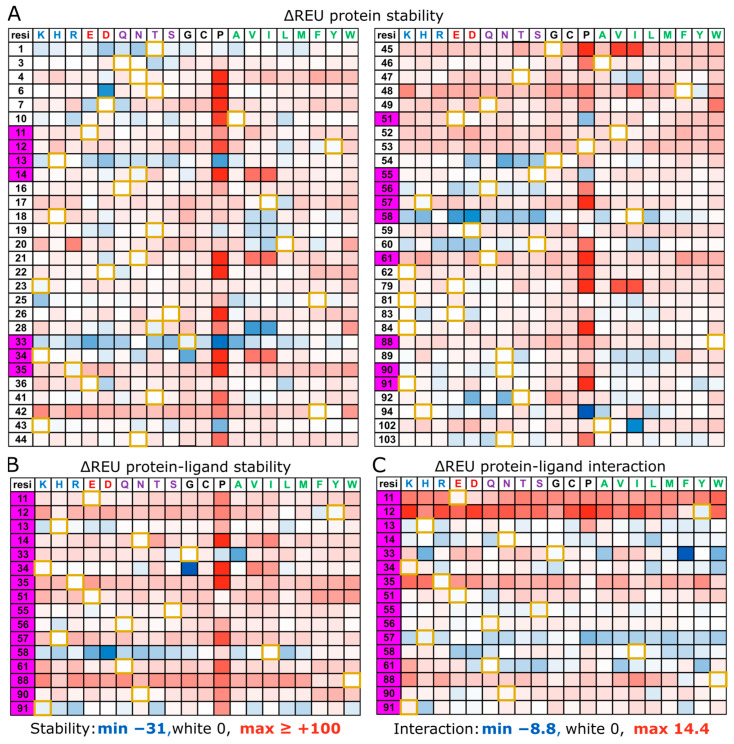
Computationally derived mutational space maps of CTB saturation mutagenesis scans on surface residues. Positions in the CTB protein are listed on the left, and mutations are shown on the top line (grouped by their structural features). Heatmaps show the ΔREU score or ΔREU interaction (compared to wild type). Control ‘mutation’ of the classical biotype CTB wt-to-wt residues are shown with gold borders, with GM1 binding residues highlighted in magenta. (**A**) A stability-based saturation mutagenesis scan of a selection of surface residues in the absence of the GM1 ligand. (**B**) A stability-based saturation mutagenesis scan of binding site residues with GM1 oligosaccharide present. (**C**) An interaction-based saturation mutagenesis scan of binding site residues with GM1 oligosaccharide present. Stability maps were capped at +100 REU (most red). Raw values can be found in Appendix A.

**Figure 4 toxins-16-00133-f004:**
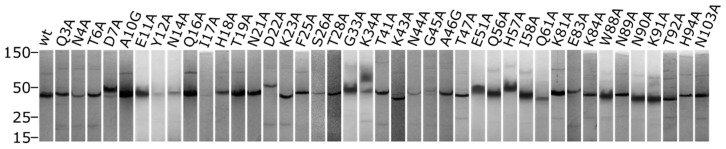
SDS-PAGE showing migration of the pentamer. Wild-type CTB runs as a pentamer when the protein sample is not boiled prior to loading it on the gel. This image presents the quantity and electrophoretic mobility of CTB pentamers that remain intact when 0.5 nmol of protein is subjected to SDS-PAGE.

**Figure 5 toxins-16-00133-f005:**
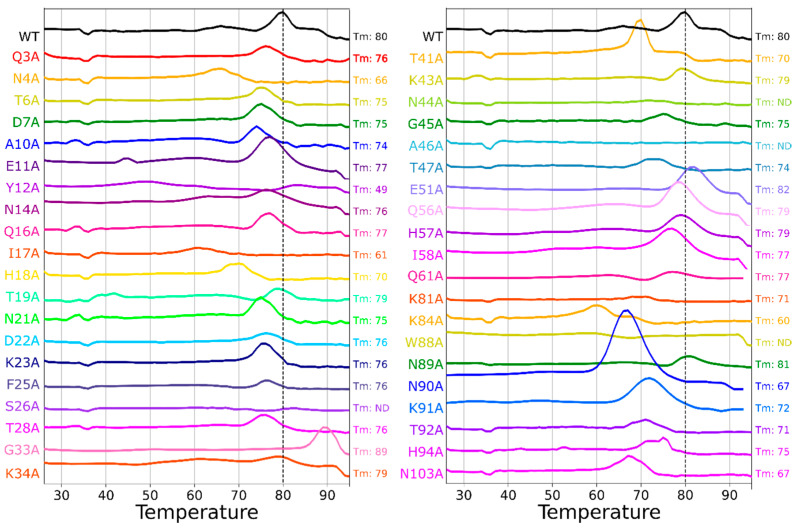
Differential scanning fluorimetry (DSF) measurements of alanine scan mutants. The melting temperature (T_m_) of WT CTB and the alanine scan mutants were determined by DSF. T_m_ could not be determined for some mutants (S26A, N44A, A46A, and W88A).

**Figure 6 toxins-16-00133-f006:**
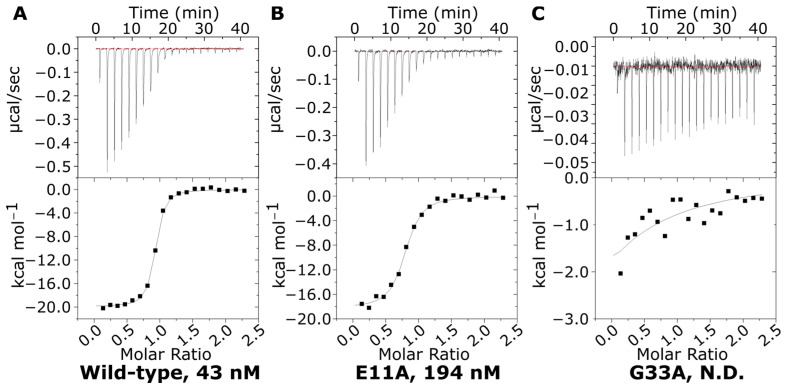
Examples of ITC data. Examples of the range of affinities observed for (**A**) the wild-type with a dissociation constant (K_d_) of 43 ± 8 nM; (**B**) E11A with a K_d_ of 194 ± 31 nM; and (**C**) G33A, whose affinity was too weak to be accurately determined. All binding curves are available in Appendix A.

**Figure 7 toxins-16-00133-f007:**
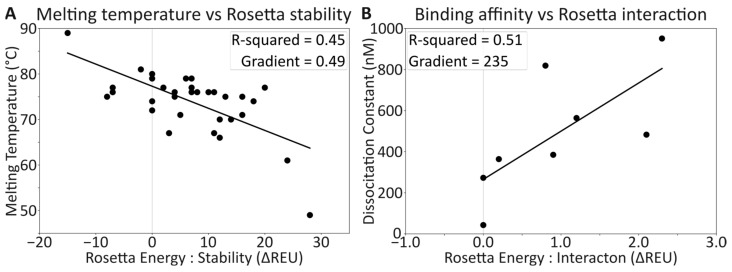
Fitting computational estimations to experimental data. (**A**) Relationship between the melting temperature as determined by DSF compared to the average Rosetta total score for 100 relaxations, showing an increase in Tm is predicted with a decrease in energy. (**B**) Relationship between the dissociation constant (K_d_) and the average Rosetta interaction score (between protein and ligand) for 100 relaxations, showing an increase in affinity with a decrease in energy. For both datasets, 10% of the datapoints were identified as outliers and removed. Original data in Appendix A.

## Data Availability

The data presented in this study are openly available at https://doi.org/10.5518/1479, [58].

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
