# Peer review of "The Mutagenic Plasticity of the Cholera Toxin B-Subunit Surface Residues: Stability and Affinity"

_toxins, 2024, doi:10.3390/toxins16030133_

Round 1

Reviewer 1 Report

Comments and Suggestions for Authors

The paper reports on computational and experimental mutagenesis of all or most surface residues comprising the pentameric cholera toxin B-subunit and the GM1 binding pocket. The results identify residues that could be changed to further stabilize the pentameric B-subunit (which already is a very stable protein); and residues that diminish assembly/stability of the pentamer and/or GM1 binding. The computational and experimental results correlate reasonably well and overall conform with the published literature. The study is technically well done and reasonably interpreted. The results are informative for protein engineering to amplify the potential utility of applying the B-subunit as a carrier of therapeutic molecules for clinical applications.

One comment: the early claim that KDEL was required for toxin entry into the ER of host cells is not supported experimentally - binding to GM1 is necessary and sufficient with some GM1 species making sorting into the retrograde pathway more efficient (see PMC2200010, PMC2200010 among others).

Author Response

Comments and Suggestions for Authors

Response from the authors are in blue

The paper reports on computational and experimental mutagenesis of all or most surface residues comprising the pentameric cholera toxin B-subunit and the GM1 binding pocket. The results identify residues that could be changed to further stabilize the pentameric B-subunit (which already is a very stable protein); and residues that diminish assembly/stability of the pentamer and/or GM1 binding. The computational and experimental results correlate reasonably well and overall conform with the published literature. The study is technically well done and reasonably interpreted. The results are informative for protein engineering to amplify the potential utility of applying the B-subunit as a carrier of therapeutic molecules for clinical applications.

We thank the reviewer for spending the time to check through our manuscript.

One comment: the early claim that KDEL was required for toxin entry into the ER of host cells is not supported experimentally - binding to GM1 is necessary and sufficient with some GM1 species making sorting into the retrograde pathway more efficient (see PMC2200010, PMC2200010 among others).

The reviewer has highlighted a paper by Lencer et al. (J. Cell. Biol. 1995, 131, 951-956) in which the authors change a KDEL sequence to LEDERAS and an RDEL sequence to RDEV and show impaired rates of toxin-induced chloride ion excretion. The reviewer is correct to comment that these toxins can reach the ER in the absence of KDEL. However, the Lencer et al. paper cites another work by Sandvig et al. (J. Cell. Biol. 1994, 126, 53-64) which shows that CTB conjugated to horseradish peroxidase and lacking the A2 sequence is only observed in the Golgi and not in the ER. In our own work we have observed that CTB localises in the medial Golgi unless the cell is flooded with CTB protein at which point it can be detected throughout the vesicular network including ER. We have amended the sentence in our revised manuscript to take into account these points (now lines 50-54)

“The Cholera toxin (CT) trafficking pathway is initiated when CTB binds to glycans presented on the cell surface [XX–XX], including ganglioside GM1 (shown in black in Figure 1C), triggering endocytosis [XX] and retrograde trafficking. CTB alone has been observed to traffic to the Golgi apparatus [https://doi.org/10.1083/jcb.126.1.53], where it accumulates in the medial compartment [XX]. The presence of a KDEL tag on the cholera toxin A2-subunit (CTA2) facilitates retrograde trafficking to the endoplasmic reticulum (ER), but mutations of this sequence also reach the ER, albeit with reduced efficiency [https://doi.org/10.1083/jcb.131.4.951].”

Reviewer 2 Report

Comments and Suggestions for Authors

 Excellent and useful study

Increased explanations would be helpful. Several additional points for comment

l13 CTB internalizes cargo to the  retrograde transport route to the Golgi/ER. this should be discussed and considered  interms of mutation-not just GM1 binding

l19  could also change intracellular routing

l34 and bacterial toxins

l36 this review does not describe the use of CTB as a carrier

l76 what about cooperativity between  B subunits in GM1 binding? Is this maintained in the mutants?

fig 1.  fig needs better explanation. What are the residue numbers on the left? is only CTB represented? which are the other CTB related toxin mutations?

l115 NB binding to GM1 oligosaccharide is lower affinity (ie distinct) from binding GM1 ganglioside.
 Was any allowance for cooperative binding made?
mutation in one subunit only accessed?

l139 How would Pro  rigidify these loops?
 and not others?

l144  apart from K34, many mutations other than Thr, are equally destabilizing

l148 H18?

l151 H18?

l152 A10?  hydrophilic..L20?

l161 K34G? H94P? R58E/D?

l171 K34G shows increase in stability with ligand

fig 3 legend -the gold labelling of the native sequence should be indicated in the legend

 fig 3B it is confusing to label this panel 'Stability' at the bottom when it is really 'instability' that is calculated

l174 the data for K94 is not shown in Fig 3B,C.

l177. binding interaction energies

205  if the mutations caused reduced pentamerization  -increased mono- dimers etc- this could be missed as pentamers are required for  nickel affinity

Author Response

Comments and Suggestions for Authors

Response from the authors are in blue

Excellent and useful study

Thank you for your comment and providing a detailed list of comments and suggestions, we will respond to each in turn.  For brevity, when appropriate, we have kept the comments quite terse, but are nonetheless very grateful for your input.

  • Increased explanations would be helpful. Several additional points for comment
  • l13 CTB internalizes cargo to the retrograde transport route to the Golgi/ER. this should be discussed and considered in terms of mutation-not just GM1 binding.

This is introduced in lines 47-52 in the introduction, we believe this is a more appropriate place for this, rather than the abstract as this study does not make estimations or observations of trafficking. We have not changed the text.

  • l19 could also change intracellular routing.

Indeed, we have added your suggestion. (now line 19)

  • l34 and bacterial toxins.

We have added an example of a non-AB5 bacterial toxin. (now line 35-36)

  • l36 this review does not describe the use of CTB as a carrier

This reference is in relation to the latter part of the sentence, considering the breadth of the AB5 toxin family.  However, we see the ambiguity and have added another reference specific to the first statement. (now line 37)

  • l76 what about cooperativity between B subunits in GM1 binding? Is this maintained in the mutants?

Cooperative binding of GM1 oligosaccharide to CTB has been known for many years, but it is actually a very small effect amounting to only a 2-fold increase in the stepwise binding constants (Schafer and Thakur, Cell Biophys., 1982, 4, 25-40; Turnbull et al., J. Am. Chem. Soc., 2004, 126, 1047-1054). While Schoen and Friere (Biochemistry, 1989, 28, 5019-5024) reported a 4-fold increase in affinity and substantial changes in stepwise enthalpy changes measured by ITC, these results are not reproducible (Turnbull et al., J. Am. Chem. Soc., 2004, 126, 1047-1054). The ITCs that we report in our current work are in line with those of Turnbull et al. in that they can be described adequately by a simple non-interacting sites binding model. We doubt that the level of signal-to-noise in our data would allow any statistically significant measurement of cooperativity. While it could be possible to investigate cooperativity in the mutants by an alternative technique such as measuring intrinsic tryptophan fluorescence, we have chosen not to investigate this as the effect of cooperativity is very minor in comparison to the overall binding affinity. We have not changed the text.

  • fig 1. fig needs better explanation. What are the residue numbers on the left? is only CTB represented? which are the other CTB related toxin mutations?

In the figure legend, we have clarified that the residue numbers are indeed residue numbers, and that only CTB is represented in panel D). The reviewer’s comment about ‘other CTB related mutations’ is presumably in reference to panel E), however, it is not practical to disentangle mutations from CTB, LTB CfxB and EcxAB as various mutations are shared across these close sequence homologs.  As their structures and sequence are so closely related, mutations are interchangeable between homologs, indeed hybrids of these proteins have been previously demonstrated.  We have added the sequence alignment use to generate the mutational-space map to the data archive for the manuscript. (changes in Fig 1 legend)

  • l115 NB binding to GM1 oligosaccharide is lower affinity (ie distinct) from binding GM1 ganglioside.

While GM1 does allow binding enhancement through multivalency at a cell membrane, the crystal structure indicates that the glucosylceramide anchor is distant from the protein surface and makes no interactions with the protein. Based on the computational method used, no differences in binding energies would be observed between GM1os and GM1 ganglioside, so GM1os was used to reduce the computational load. We have not changed the text.

  • Was any allowance for cooperative binding made?

As discussed above, the cooperativity effect is very small and it is very unlikely to be observable within the uncertainty limits of the computational calculations. Therefore, cooperative binding was not simulated/estimated in the computational evaluation, and simpler single site binding energies were estimated.  We have not changed the text.

  • mutation in one subunit only accessed?

As originally stated on lines 118-119, each mutation was made simultaneously across equivalent positions in all five chains, We have not changed the text.

  • l139 How would Pro rigidify these loops? and not others?

We have clarified that backbone geometries which natively occupy phi and psi angles complimentary to proline’s restricted backbone geometry are likely to see a stabilising effect, however those which natively possess phi and psi angles outside of prolines restricted backbone geometry are likely to find the substitution destabilising. (now line 141-145)

  • l144 apart from K34, many mutations other than Thr, are equally destabilizing

Indeed, but here we are specifically referring to mutations with side chains that branch at the beta-carbon, specifically (and only) valine, isoleucine and threonine, as this beta-carbon branching adds local steric bulk to the proximity of the backbone and restricts either the available rotameric-space of the side chain or the backbone geometry accordingly.  We have not changed the text.

l148 H18?

I did not include H18 in the list as the native residue is not beta-branched and as threonine is not particularly favourable, however I have now added it the ‘lesser extent’ comment. (now line 155)

  • l151 H18?

H18 is not one of the ‘highest energy’ residues, indeed it is one of the lowest energy options at position 18.  We have not changed the text.

  • l152 A10? .L20?

A10 only has 5 mutations which are of lower energy and L20 only has 3 mutations of lower energy. We have not changed the text.

  • l161 K34G? H94P? R58E/D?

Thank you, the K34P is a typo for K34G, we have added H94P and R58E/D (now line 167)

  • l171 K34G shows increase in stability with ligand

This is stated on line 172. We have not changed the text.

  • fig 3 legend -the gold labelling of the native sequence should be indicated in the legend

The gold boarders are already referred to in the legend on the 5th line. We have not changed the text.

  • fig 3B it is confusing to label this panel 'Stability' at the bottom when it is really ‘instability' that is calculated

These calculations are normalised against the wild-type residue at each position, where the more favourable mutation (energetically) is shaded blue, and the least favourable red.  We believe that ‘stability’ is a useful term to describe this, We have not changed the text.

  • l174 the data for K94 is not shown in Fig 3B,C.

Our apologies, this is a typo and should be K91, we have updated the script, (now line 179)

  • binding interaction energies

We have used ‘interaction’ energy consistently when referring to the computed interaction energy between GM1os and CTB (line 167 in the results and 367-373 in the methods), We have not changed the text.

  • 205 if the mutations caused reduced pentamerization  -increased mono- dimers etc- this could be missed as pentamers are required for  nickel affinity

Here we are referring to ‘pentameric stability on SDS PAGE’ specifically, as indeed, the protein is a constitutive homo-pentamer and it is thought that soluble monomers and dimers exist in fleeting concentrations.  We have observed ‘on-gel’ dissociation of the pentamers, due to the presence of the SDS and temperature of electrophoresis, We have not changed the text.

Reviewer 3 Report

Comments and Suggestions for Authors

The presented paper uses a combination of in silico and experimental approaches to investigate whether mutations at different positions in the cholera toxin B subunit will affect thermal stability or receptor binding affinity using GM1 as the receptor.

The paper is very well written and presents some results that are broadly in line with what was already known and others that were unexpected. The issue of H57A is interesting due to the correct observation that GM1 ELISA and Plasmon resonance do indeed indicate that this mutant does retain the ability to bind to GM1, but loses the ability to bind to other ligands via a secondary binding site. Thus it is able to intoxicate cells that have GM1 on the surface but not cells that do not, but that are nonetheless intoxicated by the wild type protein. It seems unlikely that there are significant differences between El Tor and classical affinities for GM1 and in any case similar studies have been done using CTB derived from classical strains including the generation of H57A mutants.

The results show that computational analysis is not yet sufficiently robust to allow mutations to be made based solely such analyses. Whereas the authors point out that many of the predicted effects were seen experimentally, there were others that were not. Also, using a scanning approach with alanine, may not give as much information as screening mutants generated by site-directed random mutagenesis. Of course this would make a computational analysis much more complicated, but would allow one to see the relative effects of the changes seen in the resulting molecules. From this point of view, would it not be useful to look at the effect of natural variations in the CTB molecule.

Lastly, it would be interesting to look at other criteria rather than thermal stability. The native molecule is highly thermostable, however, other criteria, especially for use in the context of orally administered antigens, would be a significant improvement in the acid stability. CTB dissociates at a pH significantly higher than that encountered in the stomach, and thus an improved acid-stability would be very useful. Furthermore, the human heat-labile toxin B subunit has been found to be significantly more stable and could be a starting point for comparative analysis.

The paper as it stands is broadly suitable for publication, but the preliminary nature of the approach and its applicability should be emphasized more. A last point is that CTB molecules with inserts entirely replacing the native peptides have been generated.  How might computational modelling contribute to the design of such molecules in terms of predicting stability and receptor binding affinity?

Author Response

Comments and Suggestions for Authors

Responses from the authors in blue

The presented paper uses a combination of in silico and experimental approaches to investigate whether mutations at different positions in the cholera toxin B subunit will affect thermal stability or receptor binding affinity using GM1 as the receptor.

Thank you for taking the time to review our paper.

The paper is very well written and presents some results that are broadly in line with what was already known and others that were unexpected. The issue of H57A is interesting due to the correct observation that GM1 ELISA and Plasmon resonance do indeed indicate that this mutant does retain the ability to bind to GM1, but loses the ability to bind to other ligands via a secondary binding site. Thus it is able to intoxicate cells that have GM1 on the surface but not cells that do not, but that are nonetheless intoxicated by the wild type protein. It seems unlikely that there are significant differences between El Tor and classical affinities for GM1 and in any case similar studies have been done using CTB derived from classical strains including the generation of H57A mutants.

Thank you for your comments.  Indeed, the results from H57A were intriguing, we mentioned the option of differences based on El Tor to classical variations, but my presumption is that the observed effect is driven by avidity.

The results show that computational analysis is not yet sufficiently robust to allow mutations to be made based solely such analyses. Whereas the authors point out that many of the predicted effects were seen experimentally, there were others that were not. Also, using a scanning approach with alanine, may not give as much information as screening mutants generated by site-directed random mutagenesis. Of course this would make a computational analysis much more complicated, but would allow one to see the relative effects of the changes seen in the resulting molecules. From this point of view, would it not be useful to look at the effect of natural variations in the CTB molecule.

Thank you for the comment. Indeed, the computational assessment is not deterministic, few computational estimations are, but this lack of determinism does not make these calculations obsolete, we believe these data will be important to help guide future experiments in the area.  In Figure 1E) we do present the natural variations in the CTB molecule and discuss the relationships in the discussion.

Lastly, it would be interesting to look at other criteria rather than thermal stability. The native molecule is highly thermostable, however, other criteria, especially for use in the context of orally administered antigens, would be a significant improvement in the acid stability. CTB dissociates at a pH significantly higher than that encountered in the stomach, and thus an improved acid-stability would be very useful. Furthermore, the human heat-labile toxin B subunit has been found to be significantly more stable and could be a starting point for comparative analysis.

This is a very useful suggestion for further work in the area and we have added this idea to the discussion for future work. (now lines 342-344)

The paper as it stands is broadly suitable for publication, but the preliminary nature of the approach and its applicability should be emphasized more. A last point is that CTB molecules with inserts entirely replacing the native peptides have been generated.  How might computational modelling contribute to the design of such molecules in terms of predicting stability and receptor binding affinity?

Thank you, we have added a sentence to the discussion (now line 330-333) to emphasize the limitations of the computational technique.

This is a very interesting question.  Of course, this study is looking at a selection of experimental mutations and a computation saturation scan across a large swathe of the protein.  The simplicity of these single point mutations allows for complete mutational coverage of the area of interest.  The inclusion of grafted loop sections adds orders of additional sequence complexity with each additional amino-acid in the graft and would quickly out-scale the computational resources available.  There are more modern techniques which could make grafting suggestions (for given purposes) but as this requires a contextual framework by which to generate the sequence would lack the generality we were hoping to present in this paper, and thus we consider this to be out of scope for this article.